# An unusual *endo*-selective C-H hydroarylationof norbornene by the Rh(I)-catalyzed reactionof benzamides

Kaname Shibata[1], Satoko Natsui[1], Mamoru Tobisu [1], Yoshiya Fukumoto [1] & Naoto Chatani [1]

Hydroarylation is an environmentally attractive strategy which incorporates all of the atoms contained in the substrates into the desired products. Almost all the hydroarylations of norbornene reported to date involve an *exo*-selective reaction. Here we show the *endo*-selective hydroarylation of norbornene in the Rh(I)-catalyzed reaction of aromatic amides. The addition of sterically bulky carboxylic acids enhances the *endo*-selectivity of the reaction. The results of deuterium-labeling experiments show that both the *ortho*-carbon and the *ortho*-hydrogen atoms of aromatic amides were attached to the same carbon atom of the norbornane skeleton in the hydroarylation product. These results clearly suggest that hydrometalation or carbometalation, which are commonly accepted mechanisms for the catalytic hydroarylation of C–H bonds, are not involved as the key step in the present reaction, and suggest that the reaction involves a rhodium carbene complex generated from norbornene as the key intermediate.

[1] Department of Applied Chemistry, Faculty of Engineering, Osaka University, Suita, Osaka 565-0871, Japan. Correspondence and requests for materials should be addressed to N.C. (email: chatani@chem.eng.osaka-u.ac.jp)

C atalytic addition reactions of X–Y species to alkenes are one of fundamental transformations in organic synthesis. Among the various alkenes that are used in such addition reactions, bicyclo[2.2.1]hept-2-ene (norbornene) has been extensively used in a variety of addition reactions, including hydroboration, hydrosilylation, hydroamination, carbometalation, carboesterification, and silylmetalation, because of the high reactivity of its C–C double bond due to ring strain[1]. In most cases, addition reactions of norbornene have been reported to be *exo*-selective, irrespective of the reaction mechanism, and numerous attempts have been made to explain the origin of this selectivity (Fig. 1)[2, 3].

Numerous advances in the catalytic activation of C–H bonds have been made in the past decades[4–14]. While a wide variety of functionalizations of C–H bonds has been reported to date, the hydroarylation of alkenes is the most direct and atom-economical

**Fig. 1** Conventional key steps in catalytic addition reactions to norbornene. Irrespective of the mechanism, the addition reactions of X–Y species to norbornene proceed in an *exo*-manner

| Entry | Catalyst (mol%) | T (°C)/ time(h) | 2a | 1a | Endo:exo |
|---|---|---|---|---|---|
| | | | NMRyields | | |
| 1 | [Rh(OAc)(cod)]₂ (2.5) | 160 °C/ 12 h | 84% | 3% | 6.8 : 1 |
| 2 | [RhCl(cod)]₂ (2.5) | 160 °C/ 12 h | 2% | 93% | 4.3 : 1 |
| 3 | [RhCl(cod)]₂ (2.5) / KOAc 25 mol% | 160 °C/ 12 h | 49% | 47% | 7.2 : 1 |
| 4 | [RhCl(ethylene)]₂ (2.5) / KOAc 25 mol% | 160 °C/ 12 h | 60% | 48% | 5.0 : 1 |
| 5 | RhCl(IMes)(cod)(2.5) / KOAc 25 mol% | 160 °C/ 12 h | 63% | 32% | 6.1 : 1 |
| 6 | RhCl(IPr)(cod)(2.5) / KOAc 25 mol% | 160 °C/ 12 h | 12% | 77% | 6.7 : 1 |
| 7 | [Rh(OAc)(cod)]₂ (2.5) | 160 °C/ 9 h | 77% | 10% | 7.8 : 1 |
| 8 | [Rh(OAc)(cod)]₂ (2.5) | 140 °C/ 12 h | 88% | 10% | 9.3 : 1 |
| 9 | [Rh(OAc)(cod)]₂ (1.0) | 160 °C/ 12 h | 70% | 35% | 6.4 : 1 |

**Fig. 2** *Endo*-selective hydroarylation of C–H bonds with norbornene. There are many examples of the *exo*-selective hydroarylation of C–H bonds with norbornene, but none of an *endo*-selective hydroarylation. An acetate ligand on the rhodium catalyst has a significant effect on the efficiency of the reaction

| Entry | Solvent | Acid | NMR yields | | Endo : exo |
| | | | 2b | 1b | |
| --- | --- | --- | --- | --- | --- |
| 1 | Toluene | – | 95% | 0% | 4.5 : 1 |
| 2 | Acetic acid | – | 82% | 14% | 5.7 : 1 |
| 3 | p-xylene | – | 95% | 2% | 3.9 : 1 |
| 4 | Mesitylene | – | 46% | 43% | 4.5 : 1 |
| 5 | Methylcyclohexane | – | 87% | Trace | 4.9 : 1 |
| 6 | 4-methyltetrahydropyran | – | 85% | 1% | 1.6 : 1 |
| 7 | Toluene | PivOH | 87% | 0% | 10.3 : 1 |
| 8 | Toluene | $C_6H_5COOH$ | 81% | 5% | 11.0 : 1 |
| 9 | Toluene | $2\text{-}MeC_6H_4COOH$ | 89% | Trace | 10.7 : 1 |
| 10 | Toluene | $2,6\text{-}Me_2C_6H_3COOH$ | 98% | Trace | 13.4 : 1 |
| 11 | Toluene | $2,4,6\text{-}Me_3C_6H_2COOH$ | 87% | Trace | 8.3 : 1 |
| 12 | Toluene |  | 84% | 8% | 13.1 : 1 |
| 13 | Toluene | $C_6F_5COOH$ | 93% | Trace | 4.8 : 1 |

**Fig. 3** *Endo*-selective hydroarylation of C–H bonds with norbornene. The use of bulky carboxylic acids as additives dramatically improved the *endo*-selectivity of the reaction

reaction for preparing alkylarenes, because all of the atoms of the substrates and reagents are incorporated into the desired products. There are many reports on the reaction of $C(sp^2)$–H bonds with norbornene[15–36]. Although the hydroarylation of norbornene has been extensively studied, almost all of the examples reported to date have involved an *exo*-selective reaction or the stereochemistry of the reaction products was not clearly demonstrated, irrespective of the mechanism including hydrometalation, carbometalation, heteroatom-metalation, and Friedel–Crafts type reactions. To the best of our knowledge, only a single, specific example of the *endo*-selective hydroarylation of norbornene has been reported. In that report, the reaction of mesitylene with norbornene in the presence of a W(II) carbonyl complex gave the *endo* product exclusively, however curiously, benzene, toluene, and *p*-xylene gave only *exo*-products as a single isomer, but the details of the reaction were not discussed[19].

Herein, we report on an unusual, *endo*-selective hydroarylation of norbornene that proceeds via the Rh(I)-catalyzed reaction of aromatic amides by taking advantage of an 8-aminoquinoline directing group[5]. The available evidence, based on deuterium-labeling experiments, suggests that a carbene mechanism is involved.

## Results

**Reaction development and optimization**. We initiated our study by investigating the reaction of the aromatic amide **1a** with norbornene under our previously reported hydroarylation conditions (Fig. 2)[37–40]. The reaction of amide **1a** (0.3 mmol) with norbornene (0.6 mmol) in the presence of [Rh(OAc)(cod)]$_2$ (0.0075 mmol) as the catalyst in toluene (1 mL) at 160 °C for 12 h

gave an 8.0:1 mixture of the hydroarylation product **2a** in an isolated yield of 89%. Fortunately, the major isomer was obtained in crystalline form and was recrystallized from hexane/EtOAc. Unexpectedly, an X-ray crystallographic analysis clearly showed that the major isomer of **2a** was the *endo*-isomer. Encouraged by this unusual but promising result, the effect of directing groups was examined. No reaction occurred when 2-methyl-N-(naphthalen-1-yl)benzamide (**3**) was used as the substrate. Furthermore, when quinolin-8-yl 2-methylbenzoate (**4**) and N,2-dimethyl-N-(quinolin-8-yl)benzamide (**5**) were used in place of **1a** as the substrate, no reaction took place. These results indicate that the presence of both a quinoline $N(sp^2)$ atom and a proton on the amide nitrogen is essential for the reaction to proceed. The use of 2-pyridinylmethylamine, as in the case where **6** was used as the substrate did not give the expected hydroarylation product, but, rather, a complex mixture was obtained. The reaction of N-pentafluoropheny benzamide **7** resulted in no reaction. Thus, the presence of an 8-aminoquinoline directing group is crucial for the success of the reaction.

The nature of the catalyst was next examined. Curiously, when [RhCl(cod)]$_2$ was used as the catalyst in place of [Rh(OAc)(cod)]$_2$, no hydroarylation product **2a** was produced (Fig. 2, entry 2). However, when KOAc was used as an additive, **2a** was produced in good yield (Fig. 2, entries 3 and 4), suggesting that an acetate ligand on the rhodium catalyst has a significant effect on the efficiency of the reaction. Among the rhodium complexes examined, [Rh(OAc)(cod)]$_2$ gave the best results. A shorter reaction time, lower reaction temperature, and low catalyst loading all resulted in a decreased conversion of **1a** (Fig. 2, entries 7–9).

To increase the yield of the hydroarylation product and to increase the *endo*-selectivity of the reaction, various parameters

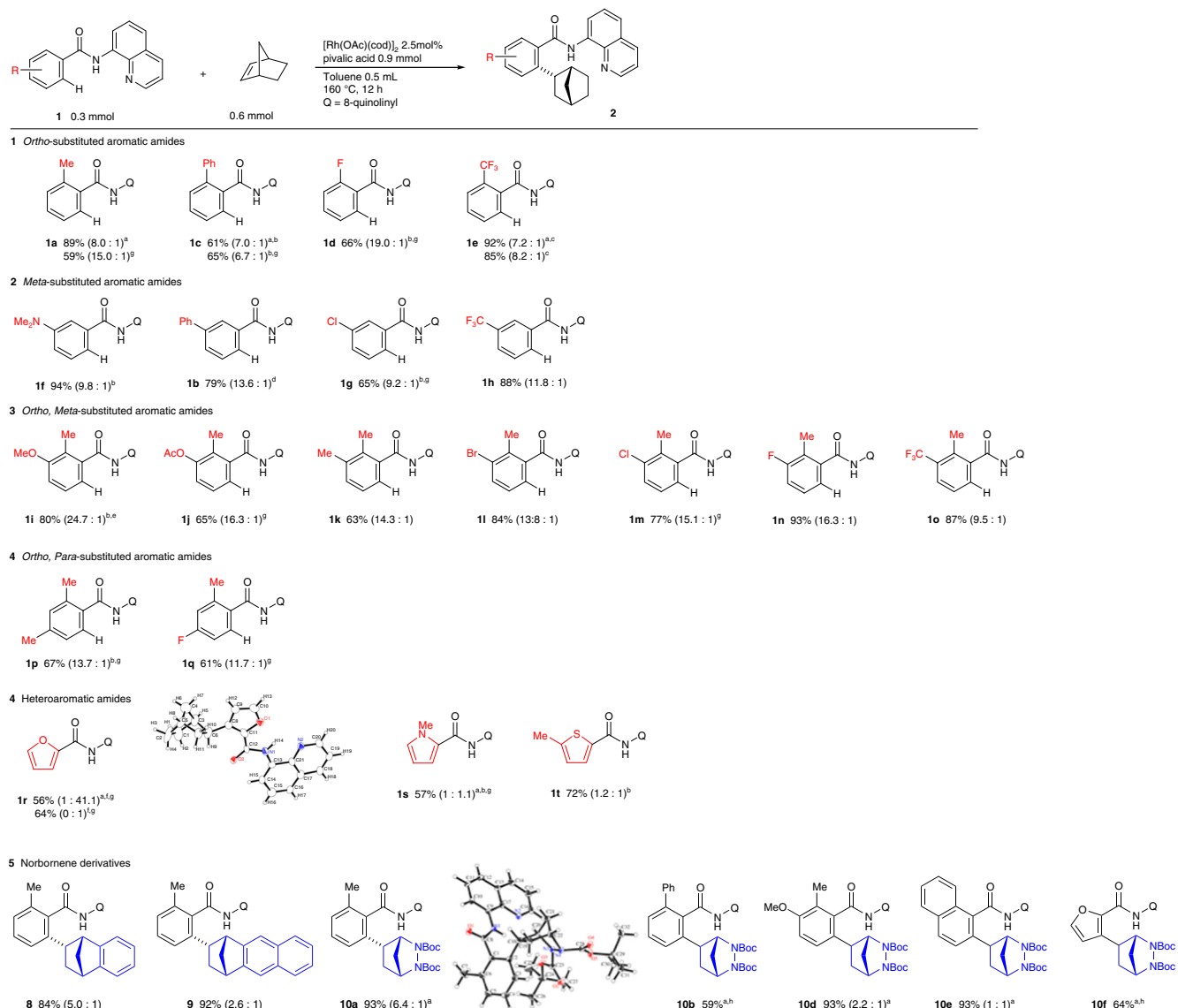

**Fig. 4** Substrate and alkene scope. [a]The reaction was carried out in the absence of pivalic acid. [b]The reaction was carried out for 24 h with catalyst (5.0 mol%). [c]The reaction was carried out for 24 h. [d]2,6-dimethylbenzoic acid was used in place of pivalic acid. [e]The reaction was carried out with pivalic acid (1 equiv) in toluene (1 mL). [f]The reaction was carried out at 120 °C. [g]Isolated by GPC. [h]The ratio could not be determined

were examined in the reaction of *meta*-phenyl-substituted amide **1b** (Fig. 3). The solvent had no significant effect on product yield (Fig. 3, entries 1–6). None of the hydrocarbon solvents examined resulted in an improved *endo*-selectivity. However, the use of 4-methyltetrahydropyrane as a solvent gave a low ratio of *endo/exo* (Fig. 3, entry 6). The addition of a carboxylic acid as an additive improved the *endo*-selectivity. It is noteworthy that the use of bulky carboxylic acids as additives dramatically improved the *endo*-selectivity: 4.5:1 for no acid, 10.3:1 for pivalic acid, 13.4:1 for 2,6-Me$_2$C$_6$H$_3$COOH (Fig. 3, entries 1, 7, and 10). Finally, the use of 3 equivalents of pivalic acid or 2,6-dimethylbenzoic acid gave the best results in terms of both conversion and *endo*-selectivity. However, trace amounts of unidentified byproducts were produced when carboxylic acids were used as additives, the formation of which frequently caused some difficulties in isolating the main products in pure form.

**Substrate scope.** The scope of amides was investigated by carrying out the reaction in the presence of 3 equivalents of pivalic acid or 2,6-dimethylbenzoic acid (Fig. 4). A number of functional groups,

including dimethylamino, methoxy, acetoxy, fluoro, bromo, and trifluoromethyl groups, were tolerated in the reaction. It was worth noting that meta-substituted aromatic amides exhibited excellent regioselectivity to give the corresponding hydroarylation products at the less-hindered C–H bonds, irrespective of the electronic nature of the substituent. Curiously, the electronic nature of the substituent also affected the *endo*-selectivity of the reaction. Thus, an electron-donating substituent tended to result in a higher *endo*-selectivity. In sharp contrast, five-membered heteroaromatic amides gave a significant amount of the *exo*-isomer. It is noteworthy that the reaction of a furancarboxamide **1r** gave the *exo*-product **2r** as a single isomer, the absolute structure of which was confirmed by X-ray crystallographic analysis. The other heteroaromatic amides gave a nearly 1:1 ratio of the hydroarylation products.

Norbornene derivatives, such as benzene-fused norbornene, naphthalene-fused norbornene, and 2,3-diazabicyclo[2.2.1]hept-5-ene were also found to participate in the present reaction to give the corresponding hydroarylation products **8–10**. The stereochemistry of the major isomer of **10a** was confirmed to be the *endo* form by X-ray crystallographic analysis.

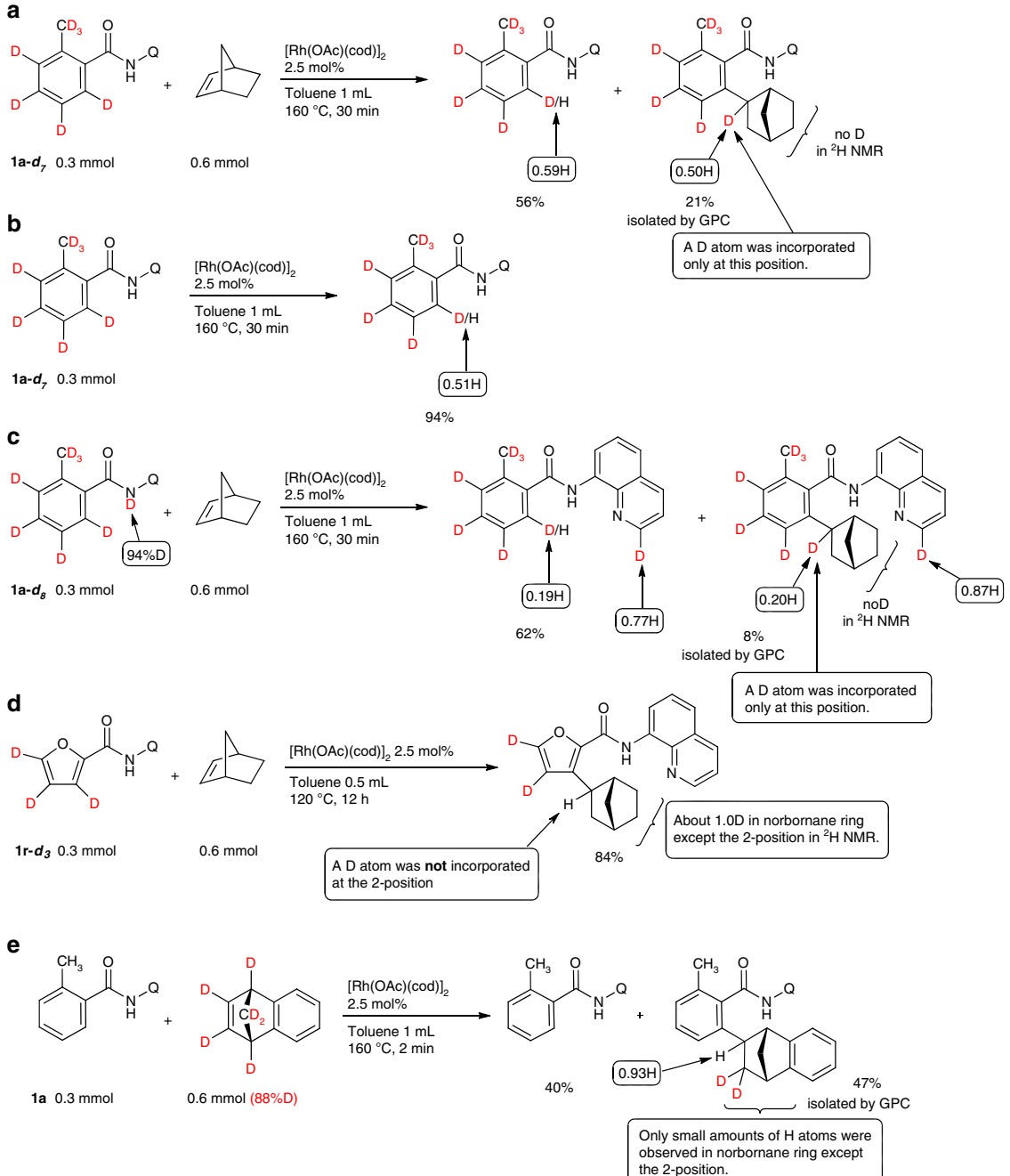

**Fig. 5** Deuterium-labeling experiments. **a**–**c** A significant amount of H/D exchange occurred in the recovered amide, and both the *ortho*-carbon and the *ortho*-H of aromatic amides were attached to the 2-position of the norbornane ring in the product. **d** However, in sharp contrast to **1a**, no deuterium incorporation was observed at the 2-position in the norbornane ring in the hydroarylation product obtained from **1r**. **e** One of the deuterium atoms migrates to the adjacent carbon

In competition experiments (Supplementary Table 1), an electron-donating group facilitated the reaction when it was carried out in the absence of a carboxylic acid. In contrast, the electronic nature of the substituent had no effect on the efficiency of the reaction in the presence of a carboxylic acid as the additive.

**Mechanistic insights**. If hydrometalation or carbometalation, commonly accepted mechanisms for catalytic hydroarylation reactions were to be involved as the key step, *exo*-selective hydroarylation would be expected to occur, as has been observed in most reported examples[15–36]. However, a high degree of *endo*-

selectivity was observed in the present system, which suggests that neither hydrometalation nor carbometalation are involved as the key step in this reaction. In an attempt to gain more insights into the mechanism of the reaction, deuterium-labeling experiments were carried out (Fig. 5). In the reaction of **1a–d7** with norbornene, a significant amount of H/D exchange occurred in the recovered amide, but only at the *ortho*-position, even when a short reaction time (30 min) was used (Fig. 5a) and it is note-worthy that a deuterium atom was incorporated only in the 2-position of the norbornane ring in the product, based on [2]H NMR spectral data. When the reaction was carried out in the absence of norbornene, a significant amount of H/D exchange was also

**Fig. 6** Proposed reaction mechanism. The generation of the rhodium carbene complex **C** and a hydride migration from the *exo*-face (**D** → **E**) are key steps

observed, but again only at the *ortho*-position (Fig. 5b). The proton source of the H/D exchange appears to be an NH bond. This H/D exchange between the *ortho*-C–H bond and the NH bond in **1a–d₇** was very rapid, making the result complicated. Because of the fast H/D exchange of the starting amide, the ratio of the deuterium atom incorporated at the 2-position of the norbornane ring is approximately 50% (0.50 H) (Fig. 5a). It is noteworthy that a deuterium atom was incorporated only in the 2-position of the norbornane ring in the product while no deuterium atoms were detected at any other position, based on²H NMR spectral data. Thus, the *ortho*-carbon and the *ortho*-hydrogen atoms of the aromatic amides attached to the same carbon atom of the norbornane skeleton in the product. The same result was also observed in the reaction of **1a–d₇** with dihydrofuran[39]. This kind of the bond connection has never observed in other systems. To make the results more clear, the reaction of **1a–d₈** was examined (Fig. 5c). As expected, the ratio of incorporated proton atoms at the *ortho*-position decreased to 19% (0.19 H) and the ratio of deuterium incorporation at the 2-position of the norbornane skeleton increased to 80% (0.20 H), but deuterium incorporation adjacent to the N(sp²) atom of the quinoline ring was also detected by ¹H NMR, indicating that protons in the quinoline ring can also serve as a proton source for the H/D exchange. It should be noted that deuterium atoms were again detected only at the 2-position in the norbornane skeleton of the hydroarylation product. In sharp contrast, different results were obtained when five-membered heterocyclic substrates were used. Thus, *exo*-products were selectively produced (Fig. 4) and no deuterium incorporation was observed in the norbornane ring of the product obtained from **1r** (Fig. 5d), suggesting that two different mechanisms are operating, depending on the structure of the substrate. Because a rapid H/D exchange in the starting amides was observed, the results obtained from deuterium-labeling experiments were complicated. To avoid such complicated results, a deuterium-labeled benzene-fused norbornene was used in attempt to develop a better understanding of the reaction

mechanism (Fig. 4). The reaction of **1a** with the deuterium-labeled benzene-fused norbornene gave the hydroarylation product **8** in which 0.93 H was observed at the 2-position of the norbornane ring, indicating that one of the deuterium atoms migrates to the adjacent carbon.

A plausible mechanism for the *endo*-selective hydroarylation is shown in Fig. 6. The coordination of the N(sp²) atom in the quinoline ring and the NH in amide **1** to a rhodium center gives the Rh(I)X species **A**. The electrophilic addition of norbornene to the rhodium complex **A** gives complex **B**, which undergoes a hydride shift to give the rhodium carbene complex **C**[41]. The oxidative addition of the *ortho*-C–H bond to the rhodium center followed by a hydride migration from the rhodium center to the carbene carbon gives **E**[42], which undergoes reductive elimination to give the hydroarylation product **2** with the regeneration of the Rh(I) species. The stereo-determining step is the hydride migration from the rhodium center to the carbene carbon in **D** (**D** → **E**), which proceeds from the *exo*-face because it is more accessible. As the alternative mechanism, the concerted oxidative addition of C–H bonds directly from **C** to **E** or the elimination of HX form **D** followed by re-addition of HX to **F** cannot be excluded. Irrespective of the mechanism, the reaction, which involves the generation of a rhodium carbene complex from an alkene, is a rare occurrence. As shown in Fig. 5a and c, when the reaction is conducted using a deuterium-labeled substrate, a deuterium atom is incorporated exclusively at the 2-position of the norbornane ring of the hydroarylation product and no deuterium atoms were detected at any other positions in the norbornane skeleton. The proposed mechanism, which involves the formation of the rhodium carbene complex **C**, is consistent with the deuterium-labeling data, although we currently have no direct experimental evidence for the generation of the rhodium carbene **C**. While diazo compounds are commonly used for the generation of metal carbenes, a metal carbene complex can also be generated from tosylhydrazones, triazoles, alkynes and cyclopropenes and these methods have been extensively used in

organic syhnthesis[43, 44]. In sharp contrast, the generation of a carbene complex from a simple alkene is very rare[45, 46]. To better understand the details of the reaction mechanism, attempts to trap the rhodium carbene complex are currently underway. In the case of five-membered heteroaromatic system, the reaction proceeds through a conventional hydrometalation or carbometalation mechanism, although we are unable to explain the difference in the mechanism at the present stage.

In summary, we report an unusual example of an *endo*-selective hydroarylation with norbornene. The use of an 8-aminoquinoline as a directing group is crucial for the success of the reaction. The addition of sterically bulky carboxylic acids was found to enhance the *endo*-selectivity of the reaction. A high degree of *endo*-selectivity was observed for a broad range of substrates with a high functional group tolerance. This reaction is a complementary method to the previously reported reaction which leads to the selective production of *exo*-isomer[15–36]. The results of deuterium-labeling experiments indicate that hydrometalation and carbometalation steps, which are commonly thought to be key steps in catalytic hydroarylation reactions reported thus far are not involved in the present reaction, instead, the generation of a carbene from norbornene is proposed. The proposed carbene mechanism is consistent with deuterium-labeling data obtained for the reaction. The most important role of a directing group in C–H functionalization is to permit the catalyst to come into close proximity to the C–H bonds, resulting in the regioselective cleavage of such bonds. However, the present results indicate that the directing group also has the potential to alter the mechanism. In this context, the design of a new directing group continues to be important in terms of developing new types of C–H functionalization that cannot currently be achieved when commonly used directing groups are used.

## Methods

**General procedure for the Rh(I)-catalyzed hydroarylation of aromatic amides with norbornene**. To an oven-dried 5 mL screw-capped vial, 3-fluoro-2-methyl-*N*-(quinolin-8-yl)benzamide (**1n**) (84 mg, 0.3 mmol), 2-norbornene (57 mg, 0.6 mmol), [Rh(OAc)(cod)]$_2$ (4.1 mg, 0.0075 mmol), pivalic acid (92 mg, 0.9 mmol) and toluene (0.5 mL) were added. The mixture was stirred for 12 h at 160 °C and then allowed to cool. The resulting mixture was filtered through a celite pad, the filtrate was washed with saturated aqueous NaHCO$_3$ (10 mL) and the organic phase concentrated in vacuo. The residue was purified by column chromatography on silica gel (eluent: hexane/EtOAc = 50/1) to afford the alkylation product **2n** (104.4 mg, 93%, *endo*:*exo* = 16.3:1) as a colorless oil.

**Data availability**. The authors declare that the data supporting the findings of this study are available within the paper and its Supplementary Information files, and also are available from the corresponding author upon reasonable request.

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

## Acknowledgements

This work was supported, in part, by a Grant-in-Aid for Scientific Research on Innovative Areas "Molecular Activation Directed toward Straightforward Synthesis" from The Ministry of Education, Culture, Sports, Science and Technology, and by JST Strategic Basic Research Programs "Advanced Catalytic Transformation Program for Carbon Utilization (ACT-C)" from Japan Science and Technology Agency (JPMJCR12YS).

## Author contributions

N.C. conceived the project and wrote the manuscript. K.S. and S.N. planned and carried out the experiments. All authors participate in the discussion of the results and commented on the manuscript.

## Additional information

**Competing interests:** The authors declare no competing financial interests.

