## [Peer Review File · Nature Communications]

Reviewer #1 (Remarks to the Author):

The authors have reported an interesting and unusual hydroarylation reaction of a strained alkene, norbornene. The surprising feature is the high endo selectivity for the addition reaction, which more typically occurs on the exo face. The authors go to some length to understand why this reaction shows this selectivity and conclude that an unexpected mechanism, facilitated by a specific ligand is responsible. They propose a rhodium carbene is formed and conduct labeling and other experiments to support this proposal. The reaction shows a reasonable scope in terms of the aryl carboxamide and more limited scope in terms of the strained alkenes that participate. In particular the level of selectivity is quite limited as the substrate is changed with several showing 1:1-2:1 endo:exo selectivity.

Overall I feel this work is suitable for Nature Communications. While the scope is limited and the utility of this rather unusual reaction is not discussed, the novelty of the transformation is likely to stimulate further work in the field. It would be valuable if some computational studies had been included to provide a bit more quantitative data in support of the carbene pathway. I wonder if the authors ever considered trying to make the carbene by an independent route to further support the Rh=C species? That might be a worthwhile experiment.

Reviewer #2 (Remarks to the Author):

This manuscript by Chatani and coworkers described a Rh(I)-catalyzed norbornene hydroarylation via amide-directed aromatic C-H activation, which led to unusual stereochemistry that suggested the involvement of Rh carbene intermediates. In particular, a catalyst system of [Rh(OAc)(cod)]₂ pre-catalyst and carboxylic acid additive was shown to promote endo-selective hydroarylation of 2-norbornenes with N-8-quinolinyl benzamides. Such unconventional stereoselectivity for norbornene hydroarylation argued against the well-established alkene hydroarylation pathways via alkene hydrometalation or arylmetalation. Mechanistic results via deuterium labeling studies supported an alternative hydroarylation pathway that involves a highly unusual formation of Rh carbene intermediates via direct interaction between 8-aminoquinoline-chelated Rh(I) center and the norbornene C=C double bond and 1,2-H shift. Subsequent carbene insertion into aromatic C-H bond, followed by Rh(III)-mediated C-C reductive elimination, generated the hydroarylation product. The scope of this hydroarylation process was demonstrated with various benzamide substrates having ortho- and meta-substituents, several heteroarene carboxylic amides, as well as several norbornene derivatives. With the notable exception of heteroaryl derivatives, most norbornene hydroarylation products were obtained in good to high endo-selectivity.

The current report is a latest example of Rh-catalyzed aromatic C-H functionalization of benzamides using 8-aminoquinoline as a doubly chelating directing group. This type of catalytic process was developed by the authors' lab and has been demonstrated with several classes of coupling partners including various alkenes and alkynes (cited in refs 40-42). It should be noted that in several of such prior reports on analogous alkene hydroarylation systems, results from deuterium-labeling supported possible involvement of H/D-migration processes that could result from Rh carbene intermediacy. In at least one of the more recent reports on hydroarylation of dihydrofurans and their lactone analogs (ref 42), the authors explicitly proposed a Rh carbene-based pathway, albeit with a different process of carbene formation (via alpha-H elimination). Thus, it's possible that the unique directing ability of 8-aminoquinoline moiety for Rh-catalyzed benzamide C-H functionalization is due to its promotion of Rh carbene intermediates from alkene-type substrates.

Considering the novelty of endo-selective norbornene hydrofunctionalization and the potential catalytic applications of late transition metal-mediated carbene complex formation from simple alkenes, the current work is an important contribution to both synthetic organic chemistry and fundamental organometallic chemistry. It is suitable for publication in Nature Communications after the following issues being addressed:

1) Regarding the connections between current study and the authors' prior reports on analogous alkene hydroarylation with N-8-quinolinyl benzamides, the authors should consider give a more detailed description on relevant results with other alkene substrates (e.g. deuterium labeling results that may suggest potential involvement of Rh carbene chemistry). In particular, the proposed Rh carbene pathway for dihydrofuran hydroarylation in ref 42 should be mentioned as important background information.

2) Ref 39 describes a Pt-catalyzed transformation of vinylsilanes that led to Pt carbene intermediates via silyl transfer. Such proposed carbene formation was based on a prior report (2015ACIE468) that the authors may either want to add as a new reference or to use to replace the current cited article for Ref 39. In fact, both reports involved carbene formation from metal vinyl complexes that are typically inaccessible with simple alkenes (in contrast to alkynes and vinylsilanes), and the authors may want to further elaborate.

3) Results summarized in Figure 3: did the authors try electron-deficient carboxylic acid additives, e.g. C₆F₅CO₂H? Compared to simple benzoic acid, it's much more electron-deficient and slightly more sterically bulky. Based on the proposed mechanism in Scheme 1, its conjugate anion (X⁻) should be more effective in promoting the key step of C-C reductive elimination that might be the rate-limiting step. The enhanced Bronsted acidity of C₆F₅CO₂H should also promote its reactivity towards Rh carbene intermediates (if such transformation is involved in the catalytic cycle).

4) Since the endo-stereoselectivity is a major point in this report, it's unfortunate that the authors did not elaborate on the effect of using bulky carboxylic acid additives on such stereochemistry. One hypothesis would be an alternative cyclometalation process towards Rh carbene formation that leads to a CNN pincer-ligated Rh(III) alkylidene intermediate (i.e. the H-X reductive elimination product from current intermediate D). Subsequent carbene reaction with carboxylic acid (HX) is expected to be syn-stereospecific and lead to stereoselective formation of intermediate E, where both groups (H and the bulky carboxylate X) add to the less shielded side of Rh=C bond, which corresponds to exo-selective C-H bond formation that places Rh-C bond at the endo-position. A similar argument about stereochemistry could possibly be made with the current proposed mechanism as shown in Scheme 1, which the authors should consider to include in the manuscript.

> *Reviewer #1 (Remarks to the Author):*

>

> *The authors have reported an interesting and unusual hydroarylation reaction of a strained alkene, norbornene. The surprising feature is the high endo selectivity for the addition reaction, which more typically occurs on the exo face. The authors go to some length to understand why this reaction shows this selectivity and conclude that an unexpected mechanism, facilitated by a specific ligand is responsible. They propose a rhodium carbene is formed and conduct labeling and other experiments to support this proposal. The reaction shows a reasonable scope in terms of the aryl carboxamide and more limited scope in terms of the strained alkenes that participate. In particular the level of selectivity is quite limited as the substrate is changed with several showing 1:1-2:1 endo:exo selectivity.*

> *Overall I feel this work is suitable for Nature Communications. While the scope is limited and the utility of this rather unusual reaction is not discussed, the novelty of the transformation is likely to stimulate further work in the field. It would be valuable if some computational studies had been included to provide a bit more quantitative data in support of the carbene pathway.*

Response:

Thank you for the positive comments. As the reviewer commented, the most important finding in the manuscript involves an unusual endo-selectivity in hydroarylation with norbornene.

Our chelation system (Rh(I)/8-aminoquinoline) was applicable to only activated alkenes, such as α,β -unsaturated esters (ref 40), styrenes (ref 41), α,β -unsaturated lactones (ref 42), dihydrofuran (ref 42), and norbornene (the present ms). However, we quite recently found new reaction conditions under which C-H alkylation with unactivated 1-alkenes unexpectedly but gratifyingly occurred (eq 1). Compared to the more extensively studied C-H alkylation with activated alkenes, alkylation with unactivated 1-alkenes continues to remain a challenging issue (for a recent review, see Dong, G. DOI: 10.1021/acs.chemrev.6b00574). Only limited examples have been reported so far. Some preliminary mechanistic studies for the equation 1 also indicate that the reaction also does not proceed via hydrometalation or carbometallation mechanisms, which are commonly accepted mechanisms for C-H alkylation with alkenes and that a carbene intermediate appears to be involved. We only recently began collaborating with a computational chemist in an attempt to elucidate the reaction mechanism for the equation 1. Because the reaction with unactivated 1-alkenes is a simple and fundamental C-H alkylation reaction with alkenes, we wish to include a DFT calculation in the manuscript for the equation 1.

At the present stage, we planned to submit the present manuscript without a DFT calculation because the major point of the present work is an unusual endo-selectivity in hydroarylation with norbornene. This surprising, interesting, and novel transformation would stimulate further investigations, not only in the field of C-H activation, but also the fields of general organic and organometallic chemistry. After we complete a DFT study of the hydroarylation with 1-alkenes, we plan to start computational studies for the present reaction, with a focus on rationalizing the endo-selectivity.

I wonder if the authors ever considered trying to make the carbene by an independent route to further support the Rh=C species? That might be a worthwhile experiment.

Response:

We made many attempts to trap the proposed rhodium carbene intermediate because, if successful, this would have given us a chance to discover a new type of transformation, based on the proposed mechanism. However, all attempts made to date have been failed. One possible explanation of the negative results is that there is an equilibrium between reactants and carbene intermediate, which is shifted in the direction of the reactants. We are currently continuing our efforts to trap a carbene intermediate to discover a new transformation.

> Reviewer #2 (Remarks to the Author):

>

> *This manuscript by Chatani and coworkers described a Rh(I)-catalyzed norbornene hydroarylation via amide-directed aromatic C-H activation, which led to unusual stereochemistry that suggested the involvement of Rh carbene intermediates. In particular, a catalyst system of [Rh(OAc)(cod)]₂ pre-catalyst and carboxylic acid additive was shown to promote endo-selective hydroarylation of 2-norbornenes with N-8-quinolinyl benzamides. Such unconventional stereoselectivity for norbornene hydroarylation argued against the well-established alkene hydroarylation pathways via alkene hydrometalation or arylmetalation. Mechanistic results via deuterium labeling studies supported an alternative hydroarylation pathway that involves a highly unusual formation of Rh carbene intermediates via direct interaction between 8-aminoquinoline-chelated Rh(I) center and the norbornene C=C double bond and 1,2-H shift. Subsequent carbene insertion into aromatic C-H bond, followed by Rh(III)-mediated C-C reductive elimination, generated the hydroarylation product. The scope of this hydroarylation process was demonstrated with various benzamide substrates having ortho- and meta-substituents, several heteroarene carboxylic amides, as well as several norbornene derivatives. With the notable exception of heteroaryl derivatives, most norbornene hydroarylation products were obtained in good to high endo-selectivity.*

The current report is a latest example of Rh-catalyzed aromatic C-H functionalization of benzamides using 8-aminoquinoline as a doubly chelating directing group. This type of catalytic process was developed by the authors's lab and has been demonstrated with several classes of coupling partners including various alkenes and alkynes (cited in refs 40-42). It should be noted that in several of such prior reports on analogous alkene hydroarylation systems, results from deuterium-labeling supported possible involvement of H/D-migration processes that could result from Rh carbene intermediacy. In at least one of the more recent reports on hydroarylation of dihydrofurans and their lactone analogs (ref 42), the authors explicitly proposed a Rh carbene-based pathway, albeit with a different process of carbene formation (via alpha-H elimination). Thus, it's possible that the unique directing ability of 8-aminoquinoline moiety for Rh-catalyzed benzamide C-H functionalization is due to

> *its promotion of Rh carbene intermediates from alkene-type substrates.*

> *Considering the novelty of endo-selective norbornene hydrofunctionalization and the potential catalytic applications of late transition metal-mediated carbene complex formation from simple alkenes, the current work is an important contribution to both synthetic organic chemistry and fundamental organometallic chemistry. It is suitable for publication in Nature Communications after the following issues being addressed:*

Thank you for the positive comments.

> *1) Regarding the connections between current study and the authors's prior reports on analogous alkene hydroarylation with N-8-quinolinyl benzamides, the authors should consider give a more detailed description on relevant results with other alkene substrates (e.g. deuterium labeling results that may suggest potential involvement of Rh carbene chemistry). In particular, the proposed Rh carbene pathway for dihydrofuran hydroarylation in ref 42 should be mentioned as important background information.*

Response:

Thank you for your suggestion. These texts have been added.

The same result was also observed in the reaction of **1a-d**₇ with dihydrofuran⁴². This kind of the bond connection has never observed in other system for C-H alkylation.

> 2) Ref 39 describes a Pt-catalyzed transformation of vinylsilanes that led to Pt carbene intermediates via silyl transfer. Such proposed carbene formation was based on a prior report (2015ACIE468) that the authors may either want to add as a new reference or to use to replace the current cited article for Ref 39. In fact, both reports involved carbene formation from metal vinyl complexes that are typically inaccessible with simple alkenes (in contrast to alkynes and vinylsilanes), and the authors may want to further elaborate.

Response:

2015ACIE468 should be 2005ACIE468. The reviewer misunderstood. In the ACIE paper, a carbene was proposed to be generated from an alkyne moiety of the starting material. There are many examples of the generation of a carbene complex from an alkyne. However, the generation of a carbene complex from an alkene is very rare. The paper cited in ref 39 is such a rare example, although a vinylsilane was used in that study.

> 3) Results summarized in Figure 3: did the authors try electron-deficient carboxylic acid additives, e.g. C₆F₅CO₂H? Compared to simple benzoic acid, it's much more electron-deficient and slightly more sterically bulky. Based on the proposed mechanism in Scheme 1, its conjugate anion (X⁻) should be more effective in promoting the key step of C-C reductive elimination that might be the rate-limiting step. The enhanced Bronsted acidity of C₆F₅CO₂H should also promote its reactivity towards Rh carbene intermediates (if such transformation is involved in the catalytic cycle).

Response:

Thank you for your suggestion. While we screened various carboxylic acids, C₆F₅CO₂H was not one of them. When C₆F₅CO₂H was used, while the product yield of **2b** was high (93% NMR yield), the endo:exo ratio was 4.8:1. The result obtained was similar to that obtained from the reaction was run in the absence of carboxylic acid additive (entry 1). This result was added in entry 13 of Figure 3.

> 4) Since the endo-stereoselectivity is a major point in this report, it's unfortunate that the authors did not elaborate on the effect of using bulky carboxylic acid additives on such stereochemistry. One hypothesis would be an alternative cyclometalation process towards Rh carbene formation that leads to a CNN pincer-ligated Rh(III) alkylidene intermediate (i.e. the H-X reductive elimination product from current intermediate D). Subsequent carbene reaction with carboxylic acid (HX) is expected to be syn-stereospecific and lead to stereoselective formation of intermediate E, where both groups (H and the bulky carboxylate X) add to the less shielded side of Rh=C bond, which corresponds to exo-selective C-H bond formation that places Rh-C bond at the endo-position. A similar argument about stereochemistry could possibly be made with the current proposed mechanism as shown in Scheme 1, which the authors should consider to include in the manuscript.

Response:

Thank you for suggesting an alternative mechanism. The alternative path has been added in Scheme 1 and the sentence related to the alternative mechanism has been changed to that shown below.

As an alternative mechanism, the concerted oxidative addition of C-H bonds directly from **C** to **E** or the elimination of HX from **D** followed by the re-addition of HX to **F** cannot be excluded.

Reviewer #2 (Remarks to the Author):

In this revised submission, the authors have sufficiently addressed referee comments from the previous review process. This manuscript is now well suited to be published in Nature Communications without further revisions.